# Impact of COVID-19 vaccination on hospitalization, hospital utilization and expenditure for COVID-19: A retrospective cohort analysis of a South African private health insured population

**Geetesh Solanki**[1,2,3]*, **Susan Cleary**[2], **Francesca Little**[4]

1 Health Systems Research Unit, South African Medical Research Council, Cape Town, South Africa,
2 Health Economics Unit, School of Public Health, University of Cape Town, Cape Town, South Africa,
3 NMG Consultants and Actuaries, Cape Town, South Africa, 4 Department of Statistical Sciences, University of Cape Town, Cape Town, South Africa

* geetesh.solanki@mrc.ac.za

## Abstract

This study quantifies the impact of COVID-19 vaccination on hospitalization for COVID-19 infection in a South African private health insurance population. This retrospective cohort study is based on the analysis of demographic and claims records for 550,332 individuals belonging to two health insurance funds between 1 March 2020 and 31 December 2022. A Cox Proportional Hazards model was used to estimate the impact of vaccination (non-vaccinated, partly vaccinated, fully vaccinated) on COVID-19 hospitalization risk; and zero-inflated negative binomial models were used to estimate the impact of vaccination on hospital utilization and hospital expenditure for COVID-19 infection, with adjustments for age, sex, comorbidities and province of residence. In comparison to the non-vaccinated, the hospitalization rate for COVID-19 was 94.51% (aHR 0.06, 95%CI 0.06, 0.07) and 93.49% (aHR 0.07, 95%CI 0.06, 0.07) lower for the partly and fully vaccinated respectively; hospital utilization was 17.70% (95% CI 24.78%, 9.95%) and 20.04% (95% CI 28.26%, 10.88%) lower; the relative risk of zero hospital days was 4.34 (95% CI 4.02, 4.68) and 18.55 (95% CI 17.12, 20.11) higher; hospital expenditure was 32.83% (95% CI 41.06%, 23.44%) and 55.29% (95% CI 61.13%, 48.57%) lower; and the relative risk of zero hospital expenditure was 4.38 (95% CI 4.06, 4.73) and 18.61 (95% CI 17.18, 20.16) higher for the partly and fully vaccinated respectively. Taken together, findings indicate that all measures of hospitalization for COVID-19 infection were significantly lower in the partly or fully vaccinated in comparison to the non-vaccinated. The use of real-world data and an aggregated level of analysis resulted in the study having several limitations. While the overall results may not be generalizable to other populations, the findings add to the evidence based on the impact of COVID-19 vaccination during the period of the pandemic.

**Data Availability Statement:** All relevant data are within the manuscript and its Supporting information files.

**Funding:** The South African Medical Research Council provided support in the form of salaries for authors Geetesh Solanki, and the University of Cape Town for Francesca Little and Susan Cleary. None of these institutions had any additional role in the study design, data collection and analysis, decision to publish, or preparation of the manuscript. The specific roles of these authors are articulated in the 'author contributions' section." Geetesh Solanki is also employed on a contractual basis by NMG Consultants and Actuaries, an independent consulting firm providing consulting and actuarial services to South African private health insurance funds. NMG provided access to the data used for the study but did not have any other role or involvement in the study design, data collection and analysis, decision to publish, or preparation of the manuscript. Geetesh Solanki's commercial affiliation to NMG therefore played no role in influencing the study.

**Competing interests:** Geetesh Solanki is employed on a contractual basis by the South African Medical Research Council and NMG Consultants and Actuaries, an independent consulting firm providing consulting and actuarial services to South African private health insurance funds. NMG provided access to the data used for the study but did not have any other role or involvement in the study design, data collection and analysis, decision to publish, or preparation of the manuscript and Geetesh Solanki's commercial affiliation to NMG therefore played no role in influencing the study. Geetesh Solanki's association with NMG does not alter our adherence to PLOS One policies on sharing data and materials.

## Introduction

The World Health Organization (WHO) declared COVID-19 a Public Health Emergency of International Concern and a pandemic on 30 January and 11 March 2020 respectively [1, 2], triggering an international response to develop a COVID-19 vaccine with four vaccine candidates entering human evaluation by March 2020 [3]. South Africa commenced its national COVID-19 vaccination program on 17 February 2021 [4]. By 31 December 2023, based on the WHO COVID-19 dashboard, South Africa had a total of 4.1 million confirmed COVID-19 cases and 70 vaccination doses per 100 population with 41 per 100 population completing the primary series and 7 per 100 population receiving at least one additional dose or booster [5]. While it is likely that COVID-19 deaths were underreported, between December 2019 and January 2022, there were 295,135 excess natural deaths in South Africa with evidence suggesting that most were due to COVID-19 [6].

While there is now substantial global evidence on the impact of COVID-19 vaccines, research in African regions remains limited. Societies in these regions have relatively poor access to health care services, varying levels of vaccine awareness and sub-optimal vaccine coverage with vaccine hesitancy emerging as an obstacle to the uptake of COVID-19 vaccines during the pandemic [7]. In South Africa, evidence confirms that the risk of COVID-19 related morbidity and mortality varied across waves and COVID-19 variants, by age, sex, level of comorbidity, and whether dependent on public or private health care [8, 9]. Studies have also confirmed the efficacy of specific COVID-19 vaccines against specific COVID-19 variants [10–13] and the effectiveness of the vaccine in "real world" situations [12, 14]. However, gaps in knowledge remain regarding the overall "real world" impact of COVID-19 vaccination as a strategy to manage COVID-19 related morbidity and health care expenditure during the pandemic.

Using Real World Data (RWD), the aim of this paper is to quantify the impact of COVID-19 vaccination on hospitalization, hospital utilization and hospital expenditure for COVID-19 infection, based on data from a South African private health insured population cohort over the entire period of the COVID-19 pandemic.

## Methods

### Study design, population and period

Strengthening and Reporting of Observational Studies in Epidemiology (STROBE) guidelines were used to conduct and report on this study [15]. The study is a retrospective cohort analysis evaluating the impact of COVID-19 vaccination on hospitalization, utilization and expenditure for COVID-19 infection. The study population included 550,332 individuals who were enrolled in one of two large health insurance funds during the study period of 1 March 2020 to 31 December 2022, corresponding to the first reported case of COVID-19 in South Africa on 5 March 2020 [16] and the end of the fourth wave and removal or COVID-19 restrictions on 30 December 2022 [17]. For individuals who were members of these insurance funds for the entire pandemic, their individual study period corresponded to the overall study period; while the actual date of entry or exit was used for those who joined or left one of the funds during the pandemic. This population represents approximately 5% of the South African private health insurance population and is comparable to the broader insured population in terms of family size, health insurance contributions and health care expenditure (S1 Table) [18].

## Exposures and outcomes of interest

The exposure of interest was defined as non-vaccinated, partly vaccinated or fully vaccinated. The South African government authorized the Johnson and Johnson (J&J) and the Pfizer BioNtech vaccines. In line with the literature [19] as well as the approach used in South Africa [20], individuals were initially categorized as non-vaccinated, becoming partly vaccinated the day after their first vaccine dose (J&J or Pfizer BioNtech), and fully vaccinated 28 days after their first J&J or 14 days after the second dose of Pfizer BioNtech. Individuals receiving booster vaccinations remained in the fully vaccinated category.

Outcomes were defined as the risk of COVID-19 hospitalization (individual having a hospital admission or not), the extent of COVID-19 hospital utilization (number of days in hospital) and the extent of COVID-19 hospital expenditure. COVID-19 hospital admissions were identified using ICD-10 code (10th revision of the International Statistical Classification of Diseases and Related Health Problems) U07.1 or U07.2 [21]. For these admissions, hospital utilization was defined using data on the date of admission and date of discharge or death in hospital. Total hospital and related expenditures took an insurer perspective and reflected the amounts paid by the health insurance funds to cover the claims submitted by health care providers for services rendered to the hospitalized individuals. These direct costs included health care providers, hospital bed days, as well as any pharmaceuticals, medical or surgical consumables dispensed. All expenditures in South African Rands (ZAR) were converted to United States Dollar ($) values using the December $/ZAR exchange rate for the given year and then inflated/deflated to be expressed in 2022 values using the South African Consumer Price Index [22].

In addition to the above exposures and outcomes, the analysis incorporated sex (male, female), age (less than 18, between 18 and 25, between 25 and 40, between 40 and 65 and greater than 65), COVID-19 comorbidities (yes, no) and province of residence (Eastern Cape, Free State, Gauteng, KwaZulu-Natal, Limpopo, Mpumalanga, North West, Northern Cape, Western Cape) as predictor variables. Comorbidities for COVID 19 considered in the analysis included cancers, chronic renal disease, congestive cardiac failure, chronic obstructive pulmonary disease, diabetes mellitus, HIV, hypercholesterolaemia, hypertension, hypothyroidism, ischaemic heart disease, pregnancy and tuberculosis, given that these have been identified as being important comorbidities for COVID-19 [23]. The individuals with these COVID-19 comorbidities were identified using Council for Medical Schemes (CMS) Guidelines for identifying these conditions using claims records [24].

## Data collection

Data for the study were extracted from the data warehouse of NMG Consultants and Actuaries (NMG), an independent consulting firm providing consulting and actuarial services to the two health insurance funds. Electronic demographic and claims records for the insured individuals, stored on a Microsoft SQL Server in the NMG data warehouse, were extracted in comma-separated values (CSV) format and then imported into STATA/SE 16.1 [25] for data cleaning, descriptive analyses, and statistical modeling.

The approach used to extract and classify the data is schematically summarized in Fig 1. From all the data for the period, a 3-step approach was used to extract and compile the data into various datasets.

In the first step, three data files were created. The first "demographic" file consisted of a record per individual who belonged to either of the two health insurance funds at any time over the study period (unique study ID, date of joining and exiting the fund, age, sex, province of residence, and comorbidities). The second "COVID-19 vaccination status" data file

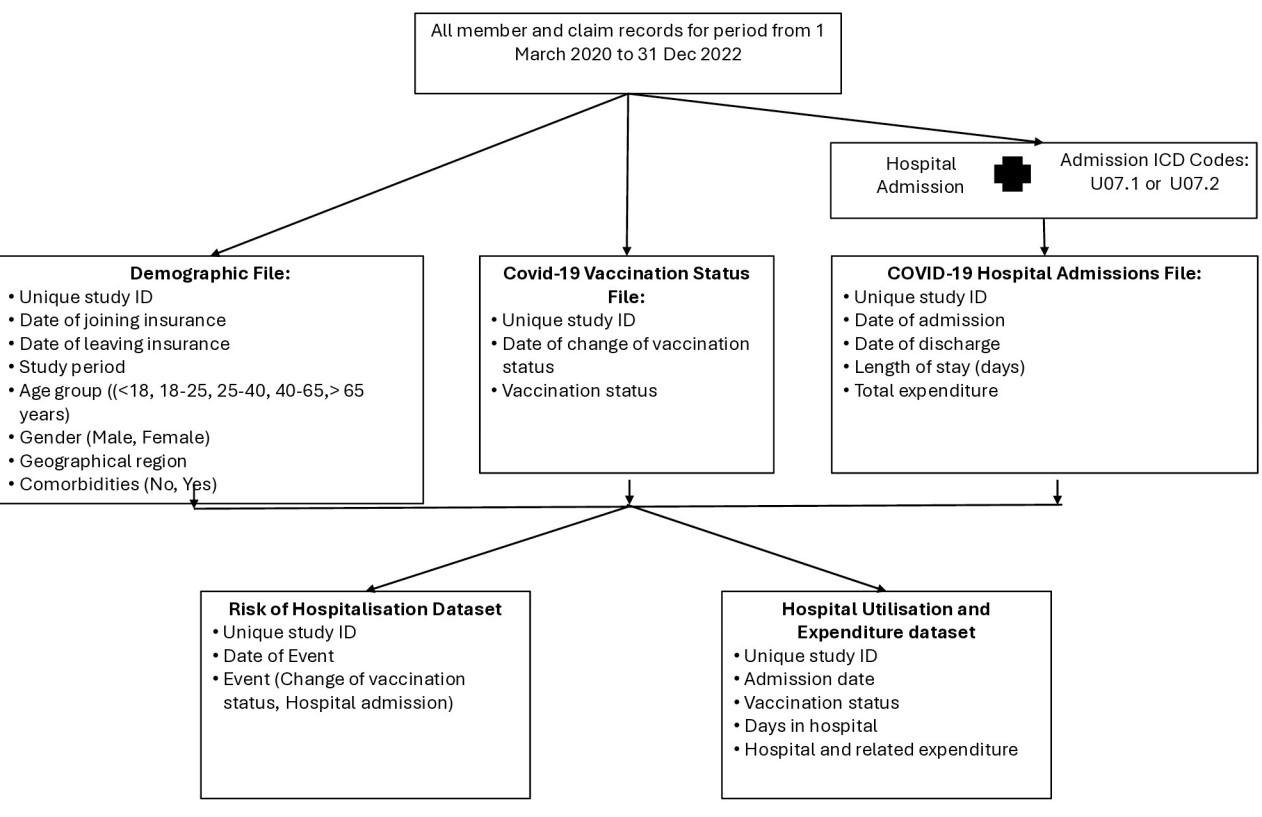

**Fig 1. Approach to data extraction.**

consisted of a record for every vaccination event in an individual (unique study ID, date of change of vaccination status, vaccination status). Individuals received their vaccinations at various points in time over the study period and their vaccination status therefore changed over time.

The "COVID-19 hospital admission" data file consisted of a record for each hospital admission for COVID-19 infection (unique study ID, date of admission, date of discharge, length of stay and total hospital and related expenditure). Individuals with multiple COVID-19 admissions had separate records for each admission. For those who were hospitalized, vaccination status (not, partly and fully vaccinated) on the date of admission was assumed.

For the second step, two separate datasets were created using information from the three files described above. Each dataset included at least one record for the 550,332 individuals included in the study population. The first dataset enabled the analysis of the risk of COVID-19 hospitalization events (S2 Table). It included "time stamped" records for every hospitalization for COVID-19 infection or COVID-19 vaccination, together with records for all individuals who had none of these events. Multiple records were created for individuals with more than one COVID-19 hospitalization or COVID-19 vaccination event. The time stamp was created by determining the number of days from study start date (1 March 2020) for the individual to the date of the event. The second dataset enabled the analysis of hospital utilization and expenditure for COVID-19 infection (S3 Table). Data included the number of days spent in hospital and expenditure per hospitalization (zero if no hospital event).

## Statistical analysis

For the analysis of the impact of vaccination on risk of hospitalization, summary statistics reported include frequencies and percentages for the number of individuals, and hospitalization frequencies and rates (number of hospital events per 1000 individuals per annum). For the multivariate analysis, survival analysis was used to model the impact of vaccination on the risk of hospitalization for COVID-19 infection to account for censoring due to no relevant hospitalization during the observation period. A Cox Proportional Hazards model was used to estimate the relative hazard of hospitalization for COVID-19 infection in partly or fully vaccinated versus non-vaccinated individuals. To factor in the censoring of the follow-period in the analysis, the Cox PH model was used for the primary analysis of the association between vaccination status and the risk of hospitalization, The model allows for those subjects who were never hospitalized during the follow-up period to be accommodated as censored observations.

Time-varying vaccination status was incorporated through repeated measures per individual relating to the periods with different vaccination statuses. The large dataset did not allow for estimation of shared frailty models. The repeated observations per individual (individuals with multiple vaccination events and multiple hospital events) was taken into consideration by incorporating the data clustering option by individual in the modelling and hence the use of robust estimators, allowing errors to be correlated within subjects but independent between subjects. The proportional hazards assumption was assessed through Kaplan-Meier curves and found to be justifiable. The association between vaccination status and risk of hospitalization are reported as unadjusted Hazard Ratios (UHR) and adjusted Hazard Ratios (AHR) with 95% confidence intervals.

For the analysis of the impact of COVID-19 vaccination on hospital utilization for COVID-19, the summary statistics included the total number of days in hospital, number of days per hospitalization event and the number of hospital days per 1000 individuals per annum. For the expenditure analysis, the summary statistics include the total expenditure in millions ($), expenditure per hospitalization event and the expenditure per individual per annum.

For the multivariate analysis of the impact of vaccination on utilization and expenditure, zero-inflated negative binomial models were used as the exploratory analyses indicated a highly skewed distribution not only because of the zeros, but also a skewed distribution on the non-zero values. The zero-inflated negative binomial model produces two sets of outputs: the count component and the zero-inflated component. The count component examines the effects of predictor variables on the expected non-zero count of the outcome variable, while the zero-inflated component explores factors influencing the probability of the presence of excess zeros. The association between vaccination and utilization and expenditure as estimated by these models are presented as percentage increase or decrease in the number of average days/ expenditures depending on whether the coefficient from the count component of the model (e.g., beta1) is positive or negative using the formula, (exp(beta1)-1) x100, and (ii) as the risk ratio for zero hospital days in the vaccinated versus the non-vaccinated, or relative to the reference group for other predictor variables, using the formula exp(beta) where beta refers to the coefficients of the zero-inflated component of the model and their 95% confidence intervals. Both unadjusted estimates from univariate models and adjusted estimates from multivariable models are presented. The lack of independence due to repeated observations per individual (individuals with multiple hospital events) was dealt with through the estimation of robust standard errors allowing correlated errors within subject and uncorrelated errors between subjects. No specific cut-off was used to determine statistical significance, and no p-values are reported. The focus is on effect sizes and their 95% confidence intervals. STATA/SE 16.1 was used for all the analyses.

### Ethical considerations

The data for the study were accessed in terms of and under the conditions set out in the agreement between NMG and the client insurance funds and between NMG and the researcher. The terms set out in these agreements ensured that the access and use of the data is in full compliance with the requirements of the Protection of Personal Information (PoPI) Act in South Africa [26]. Ethics approval for the use of the database to carry out this study was granted by the Human Research Ethics Committee of the Faculty of Health Sciences, University of Cape Town (reference number 606/2023).

## Results

### Characteristics of the study population

The study cohort included 550,332 individuals who belonged to the two health insurance funds at any time between 1 March 2020 and 31 December 2022 (1029 days). The average duration of membership for these individuals was 864 days, with 70% of the individuals being members of the funds for the full study period. At the end of the study period, 62.1% were non vaccinated, 12.2% were partly vaccinated, and 25.7% were fully vaccinated. Of all the individuals, 53.6% were female, 34.4% were less than 18 years old, 23.3% had one or more comorbidity, and the highest proportion (45.4%) of individuals resided in the Gauteng province (Table 1).

### Hospitalization risk for COVID-19 infection

As shown in Table 2, the hospitalisation rate for COVID-19 infection (events per 1000 individuals per year) was 7.45 (95%CI 7.31, 7.58) in the study population; 10.29 (95%CI 10.09, 10.49) for the non-vaccinated; 4.47 (95%CI 4.17, 4.78) for the partly vaccinated and 1.97 (95%CI 1.84, 2.11) for the fully vaccinated. Across the other predictor variables, males (7.56, 95%CI 7.36, 7.76), individuals aged 65 and above (25.62, 95%CI 24.63, 26.61), those with 1 or more comorbidity (20.12, 95%CI 19.66, 20.58) and those from the Western Cape (9.02, 95%CI 8.64, 9.40) had higher hospitalization rates.

In multivariate analysis, after adjustment for all other factors, the risk of COVID-19 hospitalization was 94.51% (aHR 0.06, 95%CI 0.06, 0.07) lower for the partly vaccinated and 93.49% (aHR 0.07, 95%CI 0.06, 0.07) lower for the fully vaccinated when compared with the non-vaccinated. The risk of hospitalization was higher for males (aHR 1.17, 95%CI 1.13, 1.12), those aged 65 and above (aHR 17.27, 95% CI 15.78, 18.89), and those with 1 or more comorbidity (aHR 3.33, 95%CI 3.18, 3.50). Individuals from all other provinces had a higher risk of COVID-19 hospitalization than individuals from the Eastern Cape, the reference group.

### Hospital utilization for COVID-19 infection

Overall hospital utilization for COVID-19 infection was 69.22 (95%CI 66.87, 71.50) days per 1000 individuals per year (Table 3). The utilization rate was 93.82 (95%CI 90.31,96.72) for the non-vaccinated, 44.11 (95%CI 37.06, 48.21) for the partly vaccinated and 20.52 (95%CI 17.25, 24.69) for the fully vaccinated. Across the other predictor variables, males (73.78, 95%CI: 70.21, 77.26), individuals aged 65 and above (322.4, 95%CI 295.21, 344.52), those with comorbidities (214.87, 95%CI 204.78, 222.94) and those from the Western Cape (86.56, 95%CI 79.41, 93.82) had the highest hospital utilization rates.

In multivariate analysis, after adjustment for all other factors, the results from the count component of the model indicated that compared with the non-vaccinated, hospital utilization was 17.70% (95% CI 24.78%, 9.95%) lower for the partly vaccinated and 20.04% (95% CI 28.26%, 10.88%) lower for the fully- vaccinated. The zero inflated component results indicated

**Table 1. Characteristics of study population.**

| Variable | Number of individuals | |
|---|---|---|
| | **N** | **%** |
| **Vaccination Status*** | | |
| Non vaccinated | 341 972 | 62.1% |
| Partly vaccinated | 66 885 | 12.2% |
| Fully vaccinated | 141 475 | 25.7% |
| **Sex** | | |
| Female | 295 116 | 53.6% |
| Male | 255 216 | 46.4% |
| **Age** | | |
| Less than 18 | 189 405 | 34.4% |
| Between 18–25 | 40 190 | 7.3% |
| Between 25–40 | 145 004 | 26.3% |
| Between 40–65 | 140 255 | 25.5% |
| Greater than 65 | 35 478 | 6.4% |
| **COVID-19 Comorbidities** | | |
| No | 422 116 | 76.7% |
| Yes | 128 216 | 23.3% |
| **Province** | | |
| Eastern Cape | 40 694 | 7.4% |
| Free State | 22 347 | 4.1% |
| Gauteng | 249 650 | 45.4% |
| KwaZulu-Natal | 85 572 | 15.5% |
| Limpopo | 19 410 | 3.5% |
| Mpumalanga | 18 514 | 3.4% |
| North West | 19 064 | 3.5% |
| Northern Cape | 10 002 | 1.8% |
| Western Cape | 85 079 | 15.5% |
| **Total** | 550 332 | 100.0% |

Vaccination Status*: Status as at end of study.

that, compared with the non-vaccinated, the relative risk of zero hospital days was 4.34 (95% CI 4.02, 4.68) for the partly vaccinated increasing to 18.55 (95% CI 17.12, 20.11) for the fully vaccinated. Across the other predictor variables, the results from the count component indicated that hospital utilization was 10.87% (95% CI 5.38%, 16.64%) higher for males compared with females, 198.39% (95% CI 156.51%, 247.10%) higher for those greater than 65-year-old compared with under 18's, and 24.82% (95% CI 17.99%, 32.05%) higher for those with comorbidities compared with those without. The zero inflated component results indicated that the relative risk of zero hospital days was 0.87 (95% CI 0.84, 0.91) for males compared with females, 0.06 (95% CI 0.06, 0.07) for those greater than 65-year-old compared with the under 18's, 0.26 (95% CI 0.25, 0.27) for those with a comorbidity compared to those without and 0.74 (95% CI 0.68, 0.82) for those from the Western Cape relative to the Eastern Cape.

## Hospital expenditure for COVID-19 infection

For the overall study population, hospital expenditure per individual for COVID-19 infection was $12.39 (95%CI $11.15, $13.61) (Table 4), with $17.85 (95%CI $16.81, $18.77) per annum

**Table 2. Univariate and multivariate analysis of impact of COVID-19 vaccination on risk of hospitalization for COVID-19 infection.**

| Variable | # of Hosp Events | | Hazard Ratio for Hospitalisation | |
|---|---|---|---|---|
| | N | Events/1000 Individuals/per annum** | UHR (95% CI) | AHR (95% CI) |
| **Vaccination Status*** | | | | |
| Non vaccinated | 9 972 | 10.29 | Reference | Reference |
| Partly vaccinated | 848 | 4.47 | 0.17 (0.15, 0.18) | 0.06 (0.06, 0.07) |
| Fully vaccinated | 791 | 1.97 | 0.20 (0.19, 0.22) | 0.07 (0.06, 0.07) |
| **Sex** | | | | |
| Female | 6 145 | 7.35 | Reference | Reference |
| Male | 5 466 | 7.56 | 1.05 (1.01, 1.09) | 1.17 (1.13, 1.22) |
| **Age** | | | | |
| Less than 18 | 927 | 1.73 | Reference | Reference |
| Between 18–25 | 271 | 2.38 | 1.33 (1.16, 1.53) | 1.99 (1.73, 2.29) |
| Between 25–40 | 2 407 | 5.86 | 3.49 (3.23, 3.78) | 5.04 (4.66, 5.46) |
| Between 40–65 | 5 431 | 13.67 | 7.64 (7.11, 8.22) | 10.32 (9.55, 11.14) |
| Greater than 65 | 2 575 | 25.62 | 13.93 (12.84, 15.11) | 17.27 (15.78, 18.89) |
| **COVID-19 Comorbidities** | | | | |
| No | 4 303 | 3.60 | Reference | Reference |
| Yes | 7 308 | 20.12 | 4.94 (4.74, 5.15) | 3.33 (3.18, 3.50) |
| **Province** | | | | |
| Eastern Cape | 827 | 7.17 | Reference | Reference |
| Free State | 461 | 7.28 | 1.08 (0.95, 1.22) | 1.26 (1.12, 1.43) |
| Gauteng | 4 812 | 6.80 | 0.95 (0.88, 1.03) | 1.33 (1.22, 1.44) |
| KwaZulu-Natal | 2 084 | 8.60 | 1.21 (1.11, 1.32) | 1.30 (1.19, 1.42) |
| Limpopo | 271 | 4.93 | 0.74 (0.64, 0.85) | 1.04 (0.90, 1.21) |
| Mpumalanga | 320 | 6.10 | 0.90 (0.78, 1.03) | 1.02 (0.88, 1.17) |
| North West | 419 | 7.76 | 1.14 (1.00, 1.30) | 1.33 (1.17, 1.52) |
| Northern Cape | 242 | 8.54 | 1.27 (1.09, 1.48) | 1.42 (1.22, 1.66) |
| Western Cape | 2 175 | 9.02 | 1.26 (1.15, 1.37) | 1.32 (1.21, 1.44) |
| **Total** | 11 611 | 7.45 | | |

Vaccination Status*: Status at time of hospitalization.

Events/1000 Individuals/per annum**: Total events/ number of individuals/study period (2.83 years) *1000

for the non-vaccinated, $6.19 (95%CI $5.20, $6.76) for the partly vaccinated, and $1.86 (95%CI $1.62, $2.18) for fully vaccinated individuals. Across the other predictor variables, males ($13.25, 95%CI $12.26, $14.23), individuals aged 65 and above ($110.17, 95%CI $109.18, $111.15), those with comorbidities ($37.57, 95%CI $36.59, $38.55) and those from the Western Cape ($16.22, 95%CI $15.23, $17.21) had the highest hospital expenditure for COVID-19 infection.

In multivariate analysis, after adjustment for all other factors, the results from the count component of the model indicated that hospital expenditure was 32.83% (95% CI 41.06%, 23.44%) lower for the partly vaccinated and 55.29% (95% CI 61.13%, 48.57%) lower for the fully vaccinated when compared with the non-vaccinated. Hospital expenditure was 9.62% (95% CI 0.37%, 18.57%) higher for males compared with females, 259.12% (95%CI 195.45%, 337.44%) higher for those aged 65 and above compared with under 18's and 19.43% (95%CI 8.29%, 31.71%) higher for those with comorbidities compared to those without. The zero inflated component results indicated that, compared with the non-vaccinated, the relative risk

**Table 3. Univariate and multivariate analysis of impact of COVID-19 vaccination on hospital utilization for COVID-19 infection.**

| | Hospital Days | | | % Difference in Hospital Days vs Reference Group*** | | Risk Ratio of Zero Hospital Day vs. Reference Group**** | |
|---|---|---|---|---|---|---|---|
| | Number of days | Days/ Event | Days/1000 individuals/ year** | Unadjusted | Adjusted | Unadjusted | Adjusted |
| **Vaccination Status*** | | | | | | | |
| Non vaccinated | 91 738 | 9.19 | 93.82 | Reference | Reference | Reference | Reference |
| Partly vaccinated | 7 976 | 9.41 | 44.11 | -0.54% (-9.26%, 9.02%) | -17.70% (-24.78%, -9.95%) | 2.29 (2.12, 2.47) | 4.34 (4.02, 4.68) |
| Fully vaccinated | 8 219 | 10.39 | 20.52 | 0.17% (-10.39%, 11.98%) | -20.04% (-28.26%, -10.88%) | 5.19 (4.80, 5.62) | 18.55 (17.12, 20.11) |
| **Sex** | | | | | | | |
| Female | 54 582 | 8.88 | 65.28 | Reference | Reference | Reference | Reference |
| Male | 53 351 | 9.76 | 73.78 | 11.90% (6.27%, 17.83%) | 10.87% (5.38%, 16.64%) | 0.98 (0.94, 1.02) | 0.87 (0.84, 0.91) |
| **Age** | | | | | | | |
| Less than 18 | 3 870 | 4.17 | 7.21 | Reference | Reference | Reference | Reference |
| Between 18–25 | 1 479 | 5.46 | 12.99 | 39.54% (6.54%, 82.77%) | 38.10% (7.47%, 77.45%) | 0.77 (0.67, 0.90) | 0.57 (0.49, 0.66) |
| Between 25–40 | 15 518 | 6.45 | 37.77 | 69.91% (45.38%, 98.58%) | 65.55% (42.17%, 92.77%) | 0.32 (0.29, 0.35) | 0.25 (0.23, 0.27) |
| Between 40–65 | 54 658 | 10.06 | 137.54 | 176.56% (138.65%, 220.50%) | 152.54% (118.19%, 192.29%) | 0.14 (0.13, 0.15) | 0.12 (0.11, 0.13) |
| Greater than 65 | 32 408 | 12.59 | 322.40 | 228.82% (182.87%, 282.24%) | 198.39% (156.51%, 247.10%) | 0.07 (0.07, 0.08) | 0.06 (0.06, 0.07) |
| **COVID-19 Comorbidities** | | | | | | | |
| No | 29 875 | 6.94 | 24.98 | Reference | Reference | Reference | Reference |
| Yes | 78 058 | 10.68 | 214.87 | 55.32% (47.18%, 63.91%) | 24.82% (17.99%, 32.05%) | 0.18 (0.17, 0.19) | 0.26 (0.25, 0.27) |
| **Province** | | | | | | | |
| Eastern Cape | 8 175 | 9.89 | 70.90 | Reference | Reference | Reference | Reference |
| Free State | 4 449 | 9.65 | 70.27 | -2.58% (-16.17%, 13.21%) | -0.18% (-14.02%, 15.89%) | 0.98 (0.86, 1.11) | 0.84 (0.74, 0.96) |
| Gauteng | 46 851 | 9.74 | 66.24 | -6.42% (-14.71%, 2.68%) | -0.88% (-9.52%, 8.60%) | 1.05 (0.96, 1.13) | 0.76 (0.70, 0.83) |
| KwaZulu-Natal | 17 369 | 8.33 | 71.64 | -19.14% (-27.01%, -10.42%) | -11.32% (-19.85%, -1.89%) | 0.81 (0.74, 0.88) | 0.74 (0.67, 0.81) |
| Limpopo | 1 981 | 7.31 | 36.02 | -28.91% (-42.39%, -12.27%) | -18.05% (-31.78%, -1.55%) | 1.40 (1.21, 1.63) | 0.97 (0.83, 1.13) |
| Mpumalanga | 2 816 | 8.80 | 53.68 | -12.15% (-27.30%, 6.15%) | -5.67% (-20.69%, 12.21%) | 1.16 (1.01, 1.33) | 0.98 (0.85, 1.13) |
| North West | 3 441 | 8.21 | 63.70 | -18.76% (-29.19%, -6.78%) | -16.46% (-26.91%, -4.53%) | 0.90 (0.79, 1.03) | 0.75 (0.66, 0.86) |
| Northern Cape | 1 986 | 8.21 | 70.08 | -18.82% (-32.45%, -2.45%) | -16.05% (-30.38%, 1.24%) | 0.81 (0.70, 0.95) | 0.70 (0.60, 0.83) |
| Western Cape | 20 865 | 9.59 | 86.56 | -6.88% (-16.40%, 3.72%) | -8.83% (-17.66%, 0.95%) | 0.78 (0.72, 0.85) | 0.74 (0.68, 0.82) |
| **Total** | 107 933 | 9.30 | 69.22 | | | | |

Vaccination Status*: Status as time of hospital event if hospitalised, and as end of study period if never hospitalised during study period

Days/1000 Individuals/per annum**: Total days/ number of individuals/study period (2.83 years) *1000

% Difference in hospital days***: Exponent of estimated coefficients (beta1) from count component output from negative binomial model—1 X 100: (exp(beta1)-1) x100.

Risk Ratio for zero hospital days****: Exponent of estimated coefficients (beta1) from zero inflated component output from negative binomial model: exp(beta1)

**Table 4. Univariate and multivariate analysis of impact of COVID-19 vaccination on hospital expenditure for COVID-19 infection.**

| | Hospital Expenditure** | | | % Difference in Hospital expenditure vs Reference Group*** | | Risk Ratio of Zero Hospital Expenditure vs. Reference Group**** | |
|---|---|---|---|---|---|---|---|
| | US Dollars (m's) | US Dollars /Event | US Dollars/ Individual/ Annum*** | Unadjusted | Adjusted | Unadjusted | Adjusted |
| **Vaccination Status*** | | | | | | | |
| Non vaccinated | 17.45 | 1 749 | 17.85 | Reference | Reference | Reference | Reference |
| Partly vaccinated | 1.12 | 1 319 | 6.19 | -25.30% (-34.09%, -15.33%) | -32.83% (-41.06%, -23.44%) | 2.28 (2.11, 2.46) | 4.38 (4.06, 4.73) |
| Fully vaccinated | 0.75 | 943 | 1.86 | -48.35% (-55.11%, -40.57%) | -55.29% (-61.13%, -48.57%) | 5.15 (4.76, 5.56) | 18.61 (17.18, 20.16) |
| **Sex** | | | | | | | |
| Female | 9.74 | 1 585 | 11.65 | Reference | Reference | Reference | Reference |
| Male | 9.58 | 1 752 | 13.25 | 10.77% (0.69%, 21.86%) | 9.75% (0.86%, 19.43%) | 0.97 (0.93, 1.01) | 0.86 (0.83, 0.90) |
| **Age** | | | | | | | |
| Less than 18 | 0.47 | 506 | 0.87 | Reference | Reference | Reference | Reference |
| Between 18–25 | 4.58 | 16 917 | 40.26 | 60.42% (2.42%, 151.26%) | 68.23% (15.25%, 145.57%) | 0.73 (0.64, 0.84) | 0.54 (0.47, 0.62) |
| Between 25–40 | 0.22 | 91 | 0.53 | 146.40% (100.00%, 203.56%) | 142.25% (99.61%, 194.00%) | 0.29 (0.27, 0.32) | 0.23 (0.21, 0.25) |
| Between 40–65 | 2.97 | 547 | 7.48 | 308.13% (237.32%, 393.80%) | 286.26% (222.83%, 362.17%) | 0.12 (0.12, 0.13) | 0.11 (0.10, 0.11) |
| Greater than 65 | 11.07 | 4 301 | 110.17 | 254.45% (189.76%, 333.59%) | 257.83% (194.09%, 335.40%) | 0.06 (0.06, 0.07) | 0.06 (0.05, 0.06) |
| **COVID-19 Comorbidities** | | | | | | | |
| No | 5.67 | 1 317 | 4.74 | Reference | Reference | Reference | Reference |
| Yes | 13.65 | 1 868 | 37.57 | 41.64% (28.51%, 56.12%) | 19.44% (8.31%, 31.72%) | 0.17 (0.16, 0.18) | 0.25 (0.24, 0.27) |
| **Province** | | | | | | | |
| Eastern Cape | 1.07 | 1 298 | 9.31 | Reference | Reference | Reference | Reference |
| Free State | 0.72 | 1 568 | 11.42 | 21.42% (-6.59%, 57.83%) | 27.82% (-3.88%, 69.96%) | 0.99 (0.87, 1.12) | 0.85 (0.75, 0.96) |
| Gauteng | 8.03 | 1 668 | 11.35 | 28.58% (8.92%, 51.79%) | 33.42% (14.57%, 55.38%) | 1.05 (0.97, 1.14) | 0.77 (0.71, 0.83) |
| KwaZulu-Natal | 3.90 | 1 869 | 16.07 | 44.15% (20.02%, 73.14%) | 49.79% (26.43%, 77.46%) | 0.83 (0.76, 0.91) | 0.75 (0.69, 0.82) |
| Limpopo | 0.25 | 914 | 4.50 | -29.88% (-51.05%, 0.45%) | -23.55% (-43.95%, 4.28%) | 1.45 (1.25, 1.68) | 0.99 (0.85, 1.15) |
| Mpumalanga | 0.41 | 1 277 | 7.79 | -1.71% (-28.12%, 34.42%) | 2.38% (-23.02%, 36.15%) | 1.17 (1.02, 1.35) | 0.99 (0.85, 1.14) |
| North West | 0.60 | 1 435 | 11.13 | 10.62% (-11.74%, 38.64%) | 4.30% (-14.95%, 27.90%) | 0.92 (0.81, 1.05) | 0.77 (0.67, 0.88) |
| Northern Cape | 0.43 | 1 774 | 15.15 | 36.96% (-5.53%, 98.55%) | 45.66% (-5.35%, 124.16%) | 0.83 (0.71, 0.98) | 0.72 (0.61, 0.85) |
| Western Cape | 3.91 | 1 798 | 16.22 | 38.38% (15.70%, 65.50%) | 38.59% (17.77%, 63.10%) | 0.79 (0.72, 0.86) | 0.75 (0.69, 0.83) |

*(Continued)*

**Table 4.** (Continued)

| | Hospital Expenditure** | | | % Difference in Hospital expenditure vs Reference Group*** | | Risk Ratio of Zero Hospital Expenditure vs. Reference Group**** | |
|---|---|---|---|---|---|---|---|
| | US Dollars (m's) | US Dollars /Event | US Dollars/ Individual/ Annum*** | Unadjusted | Adjusted | Unadjusted | Adjusted |
| Total | 19.32 | 1 664 | 12.39 | | | | |

Vaccination Status*: Status as time of hospital event if hospitalised, and as end of study period if never hospitalised during study period

Hospital Expenditure**: South African Rands converted to US Dollar terms and discounted to 2022 value

USD/Individual/per annum***: Total USD/ number of individuals/study period (2.83 years)

% Difference in hospital expenditure***: Exponent of estimated coefficients (beta1) from count component output from negative binomial model—1 X 100: (exp (beta1)-1)x100.

Risk Ratio for zero hospital expenditure****: Exponent of estimated coefficients (beta1) from zero inflated component output from negative binomial model: exp(beta1)

of zero hospital expenditure was 4.38 (95% CI 4.06, 4.73) for the partly vaccinated and 18.61 (95% CI 17.18, 20.16) for the fully vaccinated compared with the non-vaccinated. Across the other predictor variables, the results from the count component indicated that hospital expenditure was 9.75% (95% CI 0.86%, 19.43%) higher for males compared with females, 257.83% (95% CI 194.09%, 335.40%) higher for those aged greater than 65-year-old compared with the under 18's, 19.44% (8.31%, 31.72%) higher for those with comorbidities compared to those without and 38.59% (95% CI 17.77%, 63.10%) higher for those from the Western Cape compared to those from the Eastern Cape. The zero inflated component results indicated that the relative risk of zero hospital expenditure was 0.86 (95% CI 0.83, 0.90) for males compared with females, 0.06 (95% CI 0.05, 0.06) for those aged greater than 65-year-old compared with the under 18's, 0.25 (0.24, 0.27) for those with comorbidities compared to those without, and 0.75 (95% CI 0.69, 0.83) for those from the Western Cape compared with the Eastern Cape.

## Discussion

This study has estimated the impact of COVID-19 vaccination on hospitalization for COVID-19 infection based on a retrospective cohort analysis of 550,332 individuals enrolled in two private health insurance schemes during the period of the pandemic. In contrast to many studies that have reported on the efficacy of the vaccine in clinical trial situations, in population subsets using test-negative controls [27], on specific COVID-19 variants and/or vaccines, this study provides insights on the impact of COVID-19 vaccination on an aggregated basis in a "real world" population.

By the end of the study period, 12.2% of the study cohort was partly vaccinated and 25.7% was fully vaccinated. The overall hospitalization rate for COVID-19 infection was 7.45 events per 1000 individuals per annum, hospital utilization was 69.22 days per 1000 individuals per annum, and hospital expenditure was $12 per individual per annum.

Across all analyses and after adjustment for other risk factors, the study found significantly lower hospitalization events, hospital utilization and hospital expenditure for COVID-19 infection in the partly or fully vaccinated in comparison to the non-vaccinated. Each outcome will be discussed in turn.

First, the study finds that hospitalization risk for COVID-19 infection was 94.51% (aHR 0.06, 95%CI 0.06, 0.07) lower for the partly vaccinated and 93.49% (aHR 0.07, 95%CI 0.06, 0.07) lower for the fully vaccinated. These findings are in line with existing literature. A South

African study examining the impact of the J&J vaccine in health workers reported vaccine effectiveness of 67% (95% CI: 62%-71%) in preventing COVID-19-related hospitalization [12]. More broadly, at least 3 meta- analyses have reported similar results. The first, including 51 studies assessing vaccine effectiveness in the "real-world" found that COVID-19 vaccines were 97.2% (95% CI 96.1–98.3%) effective in preventing hospitalization [28]. The second, a review of 58 studies, reported a single dose of vaccine to be 66% (50%–81%) effective while two doses were 93% (89%–96%) effective in preventing hospitalization [29]. The third, a systematic review of 42 studies examining the efficacy of a range of COVID-19 vaccines in a variety of populations, concluded that the Pfizer/BioNTech vaccine had >90% effectiveness against infections requiring hospitalization after the second dose. A single dose of the J&J vaccine was >60% effective against infections requiring hospitalization [30].

Second, in terms of hospital utilization for COVID-19 infection, findings indicate that utilization was 17.70% (95% CI -24.78%, -9.95%) lower for the partly vaccinated and 20.04% (95% CI -28.26%, -10.88%) lower for the fully vaccinated; while the relative risk of zero hospital days was 4.34 (95% CI 4.02, 4.68) higher for the partly vaccinated and 18.55 (95% CI 17.12, 20.11) higher for the fully vaccinated. This compares to a reported reduction in the length of hospital stay of 38% (95% CI: 19%–52%) in a Norwegian study [31], and the 14% (95%CI -24%, 4%) reduction reported in a Chinese study [32].

Third, in terms of hospital expenditure for COVID-19 infection, findings indicate that expenditure was 32.83%(95% CI 41.06%, 23.44%) lower for the partly vaccinated and 55.29% (95% CI 61.13%, 48.57%) lower for the fully vaccinated; and the relative risk of zero hospital expenditure was 4.38 (95% CI 4.06, 4.73) higher for the partly vaccinated and 18.61 (95% CI 17.18, 20.16) higher for the fully vaccinated. In the only study we were able to find reporting on the impact of vaccination on hospital expenditure, carried out at a large academic medical center in the United States of America, vaccinated patients experienced a 26% lower cost of hospitalization compared with the non-vaccinated (P = 0.004) [33].

This study has several limitations. While the results of the study are arguably generalizable to the insured population, generalizability to the broader South African population is limited given differences in demographics, socio-economic profiles, access to care and clinical guidelines between the insured and non-insured populations. The analysis uses routine "real world" demographic and claims data collected by the health insurance funds as part of their business, not data collected for research purposes pe se. Only services and vaccines for which claims were submitted were analysed. The implications are that both COVID-19 vaccination events and hospitalization for COVID-19 infection could be over or underreported. While it is hard to predict the impact of these biases, it seems plausible that they would not negate the findings entirely, given the magnitude of the impact. In addition, the available demographic and claims data did not enable an analysis of severity of COVID-19 infection. Survival analysis or COVID-19 related mortality related assessments were also not possible due to limitations of the mortality data. Although rare, evidence now indicates concerns around safety and side-effects of COVID-19 vaccines [34, 35]. These were not taken into consideration in this study. Given that this is a retrospective observational cohort study, subject loss-to-follow up was not a concern. However, the observational nature of the study may have resulted in selection and confounding biases. Caution should therefore be exercised in interpreting the findings of this study.

## Conclusion

This study, based on a large cohort of insured individuals followed over the entire period of the pandemic, found that—all other things being equal—the risk of hospitalization, hospital

utilization and expenditure for COVID-19 infection was substantially lower amongst those who had vaccinated for COVID-19. The study provides evidence on the overall "real-world" impact of COVID-19 vaccination, particularly in insured populations and those in Sub-Saharan Africa. Given the differences in demographics, socio-economic profiles, and access to care between insured and non-insured populations, caution should be exercised in generalizing the findings of this study.

## Supporting information

**S1 Table. Comparison of study population versus rest of the insured in South Africa.** (XLSX)

**S2 Table. Hospitalisation risk dataset.** (XLSX)

**S3 Table. Hospital utilization and expenditure dataset.** (XLSX)

## Acknowledgments

The authors would like to acknowledge the institutional support to the study authors provided by the South African Medical Research Council (SAMRC). We also acknowledge NMG Consultant and Actuaries for assisting in providing access to the data, to Priya Makanjee for extracting the data, and to Jud Cornell for assistance in fine editing the script.

## Author Contributions

**Conceptualization:** Geetesh Solanki, Susan Cleary.

**Data curation:** Geetesh Solanki, Francesca Little.

**Formal analysis:** Geetesh Solanki, Francesca Little.

**Investigation:** Geetesh Solanki, Francesca Little.

**Methodology:** Geetesh Solanki, Francesca Little.

**Project administration:** Geetesh Solanki.

**Software:** Geetesh Solanki.

**Supervision:** Susan Cleary, Francesca Little.

**Validation:** Francesca Little.

**Writing – original draft:** Geetesh Solanki.

**Writing – review & editing:** Susan Cleary, Francesca Little.

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
