## [Decision Letter · Decision Letter 0]

24 Sep 2024

PONE-D-24-26424Impact of COVID-19 vaccination on COVID-19 hospitalisation, hospital related utilisation and expenditure:  Analysis of a South African private health insured population.PLOS ONE

Dear Dr. Solanki,

Thank you for submitting your manuscript to PLOS ONE. After careful consideration, we feel that it has merit but does not fully meet PLOS ONE’s publication criteria as it currently stands. Therefore, we invite you to submit a revised version of the manuscript that addresses the points raised during the review process.

We look forward to receiving your revised manuscript.

Kind regards,

Mickael Essouma, M. D.

Academic Editor

PLOS ONE

**Journal Requirements:**

Geetesh Solanki is employed on a contractual basis by NMG Consultants and Actuaries, an independent consulting firm providing consulting and actuarial services to South African private health insurance funds. 

NMG provided access to the data used for this study but did not have any additional role in the study design, data collection and analysis, decision to publish, or preparation of the manuscript. As such, there were no conflicts of interest in the conduct of the study.

We note that one or more of the authors are employed by a commercial company: NMG Consultants and Actuaries.

“The funder provided support in the form of salaries for authors, but did not have any additional role in the study design, data collection and analysis, decision to publish, or preparation of the manuscript. The specific roles of these authors are articulated in the ‘author contributions’ section.”

4. We notice that your supplementary tables are included in the manuscript file. Please remove them and upload them with the file type 'Supporting Information'. Please ensure that each Supporting Information file has a legend listed in the manuscript after the references list.

**Additional Editor Comments:**

Lines of the manuscript should be numbered to facilitate its assessment.

I highly suggest to reduce the length of the introduction. After the first paragraph, you would highlight in a second paragraph how COVID-19 vaccination issues helped (and failed to help) mitigate COVID-19-related socioeconomic consequences including morbidity and mortality during the COVID-19 pandemic, with emphasis on data from public and private health care settings in South Africa. Then, in a third paragraph, you would highlight the study's objective, including how your study would help fill gaps in knowledge about the contribution COVID-19 to mitigate socioeconomic consequences including morbidity and mortality related to COVID-19. All these elements would be reported in one page, and the other parts of the current introduction can be deleted without specific negative effect on the message you want to deliver with this manuscript. In addition to current references 25, 31 and 32, this article may be useful to build the second and third paragraphs of the introduction: https://doi.org/10.1186/s41256-022-00255-1.

Methods section: at the beginning of this section, you should report which guidelines you used to conduct and report your study (STROBE and/or 2022 CHEERS?). As highlighted by the reviewers, you should clarify the study design. When going through your manuscript, it seems to me that you conducted cross-sectional epidemiologic and economic analyses (see the attached pdf article) of prospectively collected data of the NMG cohort. Then, you should go on to clearly describe the NMG cohort (private insurance databases which included 2 large NMG databases for this analysis...), the study population (general population or specific patients [...], including inclusion and exclusion criteria) and the periods for which data have been collected. All these elements could well be recorded in a sub-section titled "Study design, period and population". In a subsequent "data collection" sub-section, you would clearly describe how you collected data of the included population, before clearly describing in a following sub-section the statistical analyses conducted, including the main parameters assessed (hospitalization, hospital utilization and expenditures). And, indeed, "Ethical considerations" would be the last sub-section of the Methods section, and that sub-section should be shorter than what is seen in the current manuscript on pages 12 and 13.

Results section: did you want to say "characteristics of the study population" where you have written "Sample description" on page 13? After this sub-section, the other sub-sections would be the three main parameters assessed in this study: hospitalization, hospital utilization and expenditures.

Discussion, conclusion and abstract sections should also be revised to fit with the results section. For the discussion specifically, follow the plan of used guidelines. Revise the figures where necessary.

Conform to PLOS ONE author guidelines. For example, the abstract should be in one block without sub-sections. Provide a more complete address for the corresponding author.

Reduce the reference count and update your references ensuring there is no citation gaming.

Extensive language editing is necessary for a better understanding of your message.

Reviewers' comments:

Reviewer's Responses to Questions

**Comments to the Author**

1. Is the manuscript technically sound, and do the data support the conclusions?

Reviewer #1: Partly

Reviewer #2: Yes

Reviewer #3: Yes

2. Has the statistical analysis been performed appropriately and rigorously? 

Reviewer #1: Yes

Reviewer #2: Yes

Reviewer #3: Yes

3. Have the authors made all data underlying the findings in their manuscript fully available?

Reviewer #1: No

Reviewer #2: No

Reviewer #3: No

4. Is the manuscript presented in an intelligible fashion and written in standard English?

Reviewer #1: Yes

Reviewer #2: Yes

Reviewer #3: No

5. Review Comments to the Author

**Reviewer #1: **Dear Authors,

This is a very important paper. Things that need more clarification

1) I assume, the hospitalisation events for vaccinated - partially or fully are after the date of vaccination. Kindly specify that.

2) If someone had more than one hospitalisation events - how were they handled?

3) For unvaccinated people - what was the date of observation if they were not hospitalised? How was the cut-off period for analyses decided. You have mentioned end of the study - what is the end? since you have described this as a cross-sectional study. Please clarify this

4) For cost - you have calculated among hospitalised only - It will be useful to have more information on the calculation of costs. You have spoked about models (negative binomial etc). Kindly provide information on the variables that were include d and how was the cost determined.

Kindly clarify the issues related to time of observation. It will help in assessment of the methods and results.

Hope these comments are useful

**Reviewer #2:** The paper is well written and the idea is well articulated. The reduction in hospitalisation and costs associated with covid hospitalisation. However the contextual information is very important, effectiveness of individual vaccines is an important piece of information which was not analysed and the cohort of relatively affluent people with access to healthcare and relatively good healthcare status. There are a few tangents in the manuscript regarding clinical case description which could detract from the strong policy message of the analyses unless it is to evaluate the effectiveness of vaccination against different severities of diseases, which would not be available since data is regarding hospitalisation, which would suppose only severe cases requiring hospitalisation would be included. The limitations also need to be acknowledged better, direct comments included

**Reviewer #3:** This manuscript presents a thorough analysis of the impact of COVID-19 vaccination on three key healthcare outcomes: hospitalization rates, hospital utilization, and healthcare expenditure. It examines a cohort of 550,332 individuals covered by private health insurance in South Africa, utilizing real-world data from March 2020 to December 2022. The study employs appropriate statistical methods, including Cox Proportional Hazards models and Zero-Inflated Negative Binomial models, to assess the effects of vaccination status on these outcomes.

Key strengths of the study include its large sample size and detailed examination of multiple outcomes. The comparative analysis of vaccinated, partly vaccinated, and unvaccinated individuals provides a clear understanding of the differential effects of partial and full vaccinations on health outcomes.

This research addresses a critical gap in the literature regarding the impact of COVID-19 vaccination on healthcare utilization and costs in low- and middle-income countries. Despite minor limitations, the manuscript offers valuable insights that can guide future public health strategies. I commend the authors for their valuable contribution.

Some areas for improvement:

Type of COVID-10 variant predominant at the time of the outcomes could influence the outcome. Vaccination type could make the findings more granular. The authors can discuss these as limitations if they could not assess these.

Also there were typos and grammatical errors limiting the overall writing quality. See additional comments in the attached document.

6. PLOS authors have the option to publish the peer review history of their article (what does this mean?). If published, this will include your full peer review and any attached files.

Reviewer #1: No

Reviewer #2: **Yes: **Usman Nasir Nakakana

Reviewer #3: No

---

## [Author Response · Author response to Decision Letter 0]

23 Oct 2024

Alll the editor and reviewer comments and our response to them are set out in the response to the reviewer note and copied below: 

Editor: 

and 

Response: We have revised the formatting of the paper to follow the guidelines. We were unable to copy the tables as required (tables exceeded the page width), but uploaded the tables in a separate excel file). 

Geetesh Solanki is employed on a contractual basis by NMG Consultants and Actuaries, an independent consulting firm providing consulting and actuarial services to South African private health insurance funds. 

NMG provided access to the data used for this study but did not have any additional role in the study design, data collection and analysis, decision to publish, or preparation of the manuscript. As such, there were no conflicts of interest in the conduct of the study.

We note that one or more of the authors are employed by a commercial company: NMG Consultants and Actuaries.

Response: We have provided an amended funding statement in the cover letter. 

“The funder provided support in the form of salaries for authors, but did not have any additional role in the study design, data collection and analysis, decision to publish, or preparation of the manuscript. The specific roles of these authors are articulated in the ‘author contributions’ section.”

Response: We have provided an amended funding statement in the cover letter. 

Response: We have provided an amended funding statement in the cover letter. 

Response: We uploaded the two final datasets used for the analysis as part of the revised submission. 

4. We notice that your supplementary tables are included in the manuscript file. Please remove them and upload them with the file type 'Supporting Information'. Please ensure that each Supporting Information file has a legend listed in the manuscript after the references list.

Response: We have removed the table as recommended. We have added legends for the two datasets that we have uploaded. 

Additional Editor Comments:

Lines of the manuscript should be numbered to facilitate its assessment.

Response: We have numbered the lines in the revised script. 

I highly suggest to reduce the length of the introduction. After the first paragraph, you would highlight in a second paragraph how COVID-19 vaccination issues helped (and failed to help) mitigate COVID-19-related socioeconomic consequences including morbidity and mortality during the COVID-19 pandemic, with emphasis on data from public and private health care settings in South Africa. 

Then, in a third paragraph, you would highlight the study's objective, including how your study would help fill gaps in knowledge about the contribution COVID-19 to mitigate socioeconomic consequences including morbidity and mortality related to COVID-19. All these elements would be reported in one page, and the other parts of the current introduction can be deleted without specific negative effect on the message you want to deliver with this manuscript. In addition to current references 25, 31 and 32, this article may be useful to build the second and third paragraphs of the introduction: https://doi.org/10.1186/s41256-022-00255-1.

Response: We agree. Although we were not able to reduce it to one page, we have reworked the introduction along the lines suggested and substantially reduced its length. 

Methods section: at the beginning of this section, you should report which guidelines you used to conduct and report your study (STROBE and/or 2022 CHEERS?). 

Response: Have indicated the guideline used (STROBE). 

As highlighted by the reviewers, you should clarify the study design. When going through your manuscript, it seems to me that you conducted cross-sectional epidemiologic and economic analyses (see the attached pdf article) of prospectively collected data of the NMG cohort. Then, you should go on to clearly describe the NMG cohort (private insurance databases which included 2 large NMG databases for this analysis...), the study population (general population or specific patients [...], including inclusion and exclusion criteria) and the periods for which data have been collected. All these elements could well be recorded in a sub-section titled "Study design, period and population". In a subsequent "data collection" sub-section, you would clearly describe how you collected data of the included population, before clearly describing in a following sub-section the statistical analyses conducted, including the main parameters assessed (hospitalization, hospital utilization and expenditures). And, indeed, "Ethical considerations" would be the last sub-section of the Methods section, and that sub-section should be shorter than what is seen in the current manuscript on pages 12 and 13.

Response: We have revised the manuscript as recommended.

Results section: did you want to say "characteristics of the study population" where you have written "Sample description" on page 13? After this sub-section, the other sub-sections would be the three main parameters assessed in this study: hospitalization, hospital utilization and expenditures.

Response: We have revised as recommended.

Discussion, conclusion and abstract sections should also be revised to fit with the results section. For the discussion specifically, follow the plan of used guidelines. Revise the figures where necessary.

Response: We have revised as recommended. 

Conform to PLOS ONE author guidelines. For example, the abstract should be in one block without sub-sections. Provide a more complete address for the corresponding author.

Response: We have revised as recommended and provided a more complete address for the corresponding author. 

Reduce the reference count and update your references ensuring there is no citation gaming.

Response: We have reduced the reference count. 

Extensive language editing is necessary for a better understanding of your message.

Response: We have had the draft re-edited. 

Reviewer #1: Dear Authors,

This is a very important paper. Things that need more clarification

1) I assume, the hospitalisation events for vaccinated - partially or fully are after the date of vaccination. 

Response: We recorded all COVID-19 related hospital events over the study period. For each event we recorded the vaccination status as at the time of the event. We have added text to the methods section to make this clearer. 

Kindly specify that.

2) If someone had more than one hospitalisation events - how were they handled?

Response: We recorded all COVID-19 related hospital events over the period and there were therefore multiple records for those who had more than one event. We have added text to make this clearer. 

3) For unvaccinated people - what was the date of observation if they were not hospitalised? How was the cut-off period for analyses decided. You have mentioned end of the study - what is the end? since you have described this as a cross-sectional study. Please clarify this

Response: We decided on a study period of 1 March 2020 to 31 December 2022 to cover the 4 waves of the pandemic in South Africa. For individuals who were members of the insurance fund for the entire period, the study start date (1 March 2020) and end date (31 December 2022) were recorded as the start and end dates respectively. For those who joined the insurance funds after 1 March 2020 or left before 31 December 2022 the actual entry date or exit dates were recorded as the study start or end dates respectively. For those who were hospitalised, the vaccination status as at the time of hospitalisation was recorded. For those not hospitalised, the vaccination status as at the end of the study period was recorded. We have amended the text in the methods section to make this clearer. 

4) For cost - you have calculated among hospitalised only - It will be useful to have more information on the calculation of costs. You have spoked about models (negative binomial etc). Kindly provide information on the variables that were included and how was the cost determined.

Response: The total hospital and related expenditures were from an insurer perspective and reflected the amounts claimed from the health insurances for services rendered to the hospitalised individuals. We have amended the text in the methods section to make this clearer. 

Kindly clarify the issues related to time of observation. It will help in assessment of the methods and results.

Response: We believe that this query has been addressed in our response to queries 1 and 3 above. 

Hope these comments are useful

Response: Yes, thank you!

Reviewer #2: 

The paper is well written and the idea is well articulated. The reduction in hospitalisation and costs associated with covid hospitalisation. However the contextual information is very important, effectiveness of individual vaccines is an important piece of information which was not analysed and the cohort of relatively affluent people with access to healthcare and relatively good healthcare status. 

There are a few tangents in the manuscript regarding clinical case description which could detract from the strong policy message of the analyses unless it is to evaluate the effectiveness of vaccination against different severities of diseases, which would not be available since data is regarding hospitalisation, which would suppose only severe cases requiring hospitalisation would be included. The limitations also need to be acknowledged better, direct comments included.

Response: We thank the reviewer for taking the time to review and comment on the paper – the comments were most useful. We have tried to address the specific comments that the reviewer raised. 

Reviewer #2 comments on paragraph original draft starting (lines 92-100) 

Evidence on privately insured populations in these regions (and globally) is also limited. With 40% of health services in the sub Saharan African region being sourced from private providers, the for-profit private sector is an important provider of health services [26]. South Africa has a highly fragmented two-tiered health system with substantial disparities in access, facilities and spending between the government-funded public health system and the private health system. Around 18% of the total South African population is covered by private health insurance[27] which is the dominant funding mechanism for the private health system. The differences in underlying demographic profiles, socio-economic differentials, and access to health services of the public and private sector-dependent populations can be expected to result in differences in their COVID-19 related outcomes. 

Reviewer comment: This is an important source of bias of the paper, the pool of subjects is defined by this access consideration and the demographic profiles vary a lot

Response: The introduction has been extensively reviewed but we have highlighted the limited generalizability of the findings in the discussion and in the abstract. 

Reviewer #2 comments on original draft lines 340-344 

“In contrast to many studies that have reported on the efficacy of the vaccine in clinical trial situations, in population sub-sets using test-negative controls[43], on specific viruses and/or vaccines, this study examined and provides insights on the impact of the COVID-19 vaccination on an aggregated basis in a “real world” population cohort of 550,332 insured individuals tracked over the entire 34 month period of the pandemic in South Africa”

Reviewer Comment: This needs to be contextualized to clarify that this is a relatively affluent cohort, with a high health access and good health status relatively

Response: We have highlighted the limited generalizability of the findings in the discussion and in the abstract. 

Reviewer #2 comments on original draft lines 424-441 

From a broader policy perspective, the responses to the COVID-19 pandemic reflected and accentuated the underlying socio-ideological and political divides within and across societies. A key fault-line in these divides was differing views on individual versus societal freedoms and responsibilities, and the role of personal responsibility[53] (any health policy that prioritises in response to factors that are posited as being under individual control - by linking either the relative payment for treatment or the extent of treatment to these factors). The findings raise questions as to whether there was a case for applying personal responsibility for COVID-19 vaccination in the study population, as implemented in some countries during the COVID-19 pandemic[54, 55]. For example, the Singapore national health insurance system required those who remained “unvaccinated by choice” after 8 December 2021 to pay for their own COVID-19 treatment on the grounds that

---

## [Decision Letter · Decision Letter 1]

14 Nov 2024

PONE-D-24-26424R1Impact of COVID-19 vaccination on COVID-19 hospitalisation, hospital related utilisation and expenditure:  A retrospective cohort analysis of a South African private health insured population.PLOS ONE

Dear Dr. Solanki,

Thank you for submitting your manuscript to PLOS ONE. After careful consideration, we feel that it has merit but does not fully meet PLOS ONE’s publication criteria as it currently stands. Therefore, we invite you to submit a revised version of the manuscript that addresses the points raised during the review process.

We look forward to receiving your revised manuscript.

Kind regards,

Mickael Essouma, M. D.

Academic Editor

PLOS ONE

Journal Requirements:

Additional Editor Comments:

The authors have substantially improved their articles. However, before it can be accepted for publication as it stands, they still need to make some amendments as described below.

ABSTRACT. The abstract contains 500 words. The abstract should be based on direct and to-the-point communication highlighting salient information from the full-text in a maximum of 300 words.

INTRODUCTION. No comment.

METHODS. Did you use printed or electronic (REDCap for example) questionnaires/data extraction sheet (keywords of the “Data collection” sub-section) to record data from patients’ insurance claim records? This information needs to be provided under the “Data collection” sub-section of the Methods section. Because “exposure” and “outcome” are keywords in the methods section of a cohort study, it is important to specify the exposure in this study (COVID-19 vaccination) in the methods section. You could do this in the first sub-section “Study design, period and population”. It is also important to have a sub-section (for example between the “Data collection” and “Statistical analysis” sub-sections) “Outcomes of interest” (in the Methods) where you should clearly state the outcomes in your study: hospitalisation, hospital utilisation, and spending for COVID-19 infection. So, the text on lines 178-181 of the current manuscript would better be edited as proposed here in the “Outcomes of interest” sub-section. For clarity purposes, it would be better to specify “COVID-19 comorbidities” rather than just write “comorbidities” on lines 137, 139 and 140. This means the “Statistical analysis” sub-section of the Methods section should be further edited in a way that clearly highlights: (i) whether you exported data from a software to the statistical analysis software used, (ii) a complete records of softwares used for statistical analysis, (iii) the statistical analyses carried out (assessment of the effect of COVID-19 vaccination on study outcomes), clear description of statistical tests used for that purpose, steps observed during those statistical analyses, and how did you consider test results to be statistically significant, and finally (iv) how you report results in the article (including effect magnitude estimates such as hazard ratios, rates [with 95% confidence intervals], figures, tables…). Along these lines, should the sentence “A Cox Proportional Hazards model was used to estimate the relative hazard of hospitalisation” not be edited to read “A cox proportional Hazards model was used to estimate the hazard for hospitalisation in vaccinated versus non-vaccinated subjects.” (doi:10.1097/EDE.0b013e3181c1ea43.)? Did you make every effort to consider losses-to-follow up in statistical analyses? The text on lines 207 through 223 should be in a single paragraph because it is centred around a single idea. In other words, it is immediately after the sentence on lines 207-209 (that should be edited for simplicity and more clarity) that you should explain what the zero-inflated negative binomial model produces as you did on lines 214-223, but in a more concise way. These comments also make lead me to observe that the manuscript’s title (and therefore also keywords) needs to be edited in these likes for clarity purpose: “Impact of COVID-19 vaccination on hospitalisation, hospital utilisation and expenditure for COVID-19: a retrospective cohort analysis of a South African private health insured population”. Sentences addressing study outcomes should also be edited like this throughout the article. Is it "Partly vaccinated" or "partially vaccinated" throughout the manuscript (including in Tables)?

RESULTS. Consider adding 95% confidence interval to rates reported in the results section (at least in the full-text manuscript). The text on lines 235-240 fails to mention the mean/median duration of follow-up of this retrospective cohort study. What was the proportion of loss-to-follow up? Can you complete the footnotes of Table 1 with some information on "chronic conditions" (which chronic conditions did you observe, at least the major ones? This is especially important because those chronic conditions are also predictors of COVID-19 related hospitalisations.)? It is interesting to report in the manuscript why you chose Eastern Cape as the province of reference for the statistical analysis of the association between province and COVID-19 related hospitalisation? In the footnotes of Table 2, is it not “status at time of hospitalisation” rather than “Status at tim,e of hospital event”? Should the title of Table 2 not be something like “Results of univariate and multivariate analysis of predictive factors of hospitalisation for COVID-19”? Is this punctuation “?” necessary after the phrase “Chronic condition”. And once more, it would be interesting to know the chronic conditions that were associated with hospitalisation for COVID-19. Would it not be better to replace “chronic conditions” by “Comorbidities” in Table 2 and “COVID-19 comorbidities” in Table 1? I think it would be more right to write “hospitalisation for COVID-19” rather than COVID-19 related hospitalisation in Table 2 and throughout the manuscript. Same remarks for Tables 3, 4, S2 and S3. Should the title of Table 1 not be “Characteristics of the study population”? Is it not "versus the rest of the insured" (instead of "versus rest of the rest of the insured" in the title of Table S1? Should it not read “Hospital utilisation for COVID-19 infection” on line 263? Same remark for the sub-title on line 288. Similarly, should it not be “The overall rate of hospital utilisation for COVID-19 infection was…” on line 264? Same remark for the sentence on line 289. It would be better to report the costs in US dollars like this: $ the amount and then US, e.g., $12 US

DISCUSSION. Line 321: 7.45 per? Caution should be exercised when choosing words and interpreting findings in the discussion. For example, did you really assess risk factors (doi:10.1097/EDE.0b013e3181c1ea43.)? To avoid creating confusion in the discussion section, I would advise to start with a statement of main study findings (as you did) in a paragraph, discuss each of the assessed ouctomes of interest (hospitalisation due to COVID-19 infection, hospital utilization due to COVID-19 infection, and direct costs due to COVID-19 infection) in a separate paragraph (shorter than current paragraphs), and finally state study strengths, limitations and implications (for policies, clinical practice, and future research using the same insurance records and other records) in one to two paragraphs. When stating study limitations in the discussion section, it is very important to address the generalizability of your findings (this is a reason why you should provide us with 95% confidence intervals of the estimated rates; so, the text on lines 399-403 should be in the limitations statement.) and unavoidable study limitations related to the study design (cohort in this case; so what about selection bias, unmeasured confounding? What about the effect of losses to follow up in study results? Reverse causation? Why did you choose hazard ratios rather than relative risks as effect magnitude estimates [doi:10.1097/EDE.0b013e3181c1ea43 and https://doi.org/10.1016/j.jclinepi.2021.09.016]?) and the timeline of data collection (retrospective in this study: so selection bias, incomplete data collection/recall bias and inability to segregate data [for example by type of COVID-19 vaccine received] expected). Line 340: “in most other studies”. Which ones (you mentioned in the introduction that there is a lack of studies similar to this one)? Same remark on line 344 (“other studies”). And indeed, you do not report any effect magnitude estimate (RR, HR or even OR) from any of the studies cited on lines 343-360. If there are no studies comparable to yours (in terms of methodology) with effect magnitude estimates but only incidence estimates for hospitalization due to COVID-19 after exposure to COVID-19 vaccination, then discuss incidence rates and call for further studies to further assess effect magnitude estimates of hospitalisation due to COVID-19 infection after exposure to COVID-19 vaccination. Not even a small comment about patient survival when you used a Cox proportional Hazards model (why was it not even a an (secondary at least?) outcome of interest in the first place? I guess patients were hospitalised for severe/critical COVID-19 and may be hospital utilisation for COVID-19 infection included hospital utilisation for COVID-19 cases not hospitalised (those with mild to moderate COVID_19 infection) and COVID-19 cases (those with severe/critical COVID-19 infection). However, there is no such specification in any part of the article not even in the discussion when addressing limitations. If the information on COVID-19 severity is available, it would also be great to specify which COVID-19 infection severity classification scheme was used in your study. Same remark for expenditure for COVID-19. I guess you are addressing direct costs in this article. Can you detail what was included in that spending (at least in a table uploaded as a supplemental material)? These bits of information will increase the validity and strength of your findings and interpretations. The discussion section always requires a great editing effort so as to be as exhaustive as possible using as few words as possible.

CONCLUSION. It should be revised to provide a direct and to-the-point communication on the main results readers should keep in mind when reading your study, as well as the most important research perspective for future studies on this topic. I advise avoiding strong words such as “confirm” because this is not an adjudicating data synthesis.

DECLARATIONS. I do not see them (e.g., Conflicts of interests, Author contributions, and Acknowledgments after the CONCLUSION section and before “REFERENCES”).

REFERENCES. I am unable to open the link to reference 17. So, make sure all the links provided work well.

Reviewers' comments:

Reviewer's Responses to Questions

**Comments to the Author**

1. If the authors have adequately addressed your comments raised in a previous round of review and you feel that this manuscript is now acceptable for publication, you may indicate that here to bypass the “Comments to the Author” section, enter your conflict of interest statement in the “Confidential to Editor” section, and submit your "Accept" recommendation.

Reviewer #3: All comments have been addressed

2. Is the manuscript technically sound, and do the data support the conclusions?

Reviewer #3: Yes

3. Has the statistical analysis been performed appropriately and rigorously? 

Reviewer #3: Yes

4. Have the authors made all data underlying the findings in their manuscript fully available?

Reviewer #3: Yes

5. Is the manuscript presented in an intelligible fashion and written in standard English?

Reviewer #3: Yes

6. Review Comments to the Author

Reviewer #3: (No Response)

7. PLOS authors have the option to publish the peer review history of their article (what does this mean?). If published, this will include your full peer review and any attached files.

Reviewer #3: No

---

## [Author Response · Author response to Decision Letter 1]

31 Dec 2024

Mickael Essouma, M. D.

Academic Editor

PLOS ONE

Dear Mickael

Response to reviewer comments on Ref.: PONE-D-24-26424

Impact of COVID-19 vaccination on COVID-19 hospitalisation, hospital related utilisation and expenditure: Analysis of a South African private health insured population. 

Thank you for the reviewer comments on our manuscript. We have revised the manuscript and summarise our response to the comments below. 

Additional Editor Comments:

The authors have substantially improved their articles. However, before it can be accepted for publication as it stands, they still need to make some amendments as described below.

ABSTRACT. The abstract contains 500 words. The abstract should be based on direct and to-the-point communication highlighting salient information from the full-text in a maximum of 300 words.

Response: We have reduced the abstract word count to 290 words. 

INTRODUCTION. No comment.

METHODS. Did you use printed or electronic (REDCap for example) questionnaires/data extraction sheet (keywords of the “Data collection” sub-section) to record data from patients’ insurance claim records? This information needs to be provided under the “Data collection” sub-section of the Methods section. 

Response: We have added the following sentence under data collection to provide greater clarity on the data capture process and formats. “Electronic demographic and claims records for the insured individuals, stored on a Microsoft SQL Server in the NMG data warehouse, were extracted in comma-separated values (CSV) format and then imported into STATA/SE 16.1for data cleaning, descriptive analyses, and statistical modeling.”

Because “exposure” and “outcome” are keywords in the methods section of a cohort study, it is important to specify the exposure in this study (COVID-19 vaccination) in the methods section. You could do this in the first sub-section “Study design, period and population”. It is also important to have a sub-section (for example between the “Data collection” and “Statistical analysis” sub-sections) “Outcomes of interest” (in the Methods) where you should clearly state the outcomes in your study: hospitalisation, hospital utilisation, and spending for COVID-19 infection. So, the text on lines 178-181 of the current manuscript would better be edited as proposed here in the “Outcomes of interest” sub-section. 

Response: We have added a “Exposure and outcomes of Interest” section and moved lines 178-186 (earlier draft) to the section. In the section, we have added wording to more clearly specify the exposures and outcomes of interest for the study 

For clarity purposes, it would be better to specify “COVID-19 comorbidities” rather than just write “comorbidities” on lines 137, 139 and 140. 

Response: We have specified COVID-19 comorbidities as recommended in the “Exposures and outcomes interest” section 

This means the “Statistical analysis” sub-section of the Methods section should be further edited in a way that clearly highlights: 

(i) whether you exported data from a software to the statistical analysis software used, 

(ii) complete records of softwares used for statistical analysis, 

Response: We have added wording to address this under the data collection section. 

(iii) the statistical analyses carried out (assessment of the effect of COVID-19 vaccination on study outcomes), clear description of statistical tests used for that purpose, steps observed during those statistical analyses, and how did you consider test results to be statistically significant, and finally how you report results in the article (including effect magnitude estimates such as hazard ratios, rates [with 95% confidence intervals], figures, tables…). Along these lines, should the sentence “A Cox Proportional Hazards model was used to estimate the relative hazard of hospitalisation” not be edited to read “A cox proportional Hazards model was used to estimate the hazard for hospitalisation in vaccinated versus non-vaccinated subjects.” (doi:10.1097/EDE.0b013e3181c1ea43.)? 

Response: We have edited the statistical methods section to further clarify the methods used for analysis and reporting. We have added wording to clarify how effect sizes estimated from the models are reported in the paper.We have revised the “A Cox Proportional Hazard model…” sentence as suggested. On the matter of statistical significance, we have added wording to indicate that “No specific cut-off was used to determine statistical significance, and no p-values are reported. The focus is on effect sizes and their 95% confidence intervals.”

(iv) Did you make every effort to consider losses-to-follow up in statistical analyses? 

Response: In Statistical analysis section we have added the following wording: “To factor in the censoring of the follow-period in the analysis, the Cox PH model was used for the primary analysis of the association between vaccination status and the risk of hospitalization. The model allows for those subjects who were never hospitalized during the follow-up period to be accommodated as censored observations.” 

In the discussion section, we have added the following sentence: “Given that this is a retrospective observational cohort study, subject loss-to-follow up was not a concern. However, the observational nature of the study may have resulted in selection and confounding biases. Caution should therefore be exercised in interpreting the findings of this study.”

The text on lines 207 through 223 should be in a single paragraph because it is centred around a single idea. In other words, it is immediately after the sentence on lines 207-209 (that should be edited for simplicity and more clarity) that you should explain what the zero-inflated negative binomial model produces as you did on lines 214-223, but in a more concise way. 

Response: We have removed the break to consolidate the wording into a paragraph. We have also added further details describing how the effect sizes for the association between predictor variables were determined from these models and reported in the manuscript.

These comments also make lead me to observe that the manuscript’s title (and therefore also keywords) needs to be edited in these likes for clarity purpose: “Impact of COVID-19 vaccination on hospitalisation, hospital utilisation and expenditure for COVID-19: a retrospective cohort analysis of a South African private health insured population”. Sentences addressing study outcomes should also be edited like this throughout the article. 

Response: We have re-worded the manuscript title and text as suggested. We have also added to the keyword list. 

Is it "Partly vaccinated" or "partially vaccinated" throughout the manuscript (including in Tables)?

Response: We reworded all instances of partially vaccinated to partly vaccinated to ensure consistency. 

RESULTS. 

Consider adding 95% confidence interval to rates reported in the results section (at least in the full-text manuscript). 

Response: We have added the 95% confidence intervals for the unadjusted rates in the text. Adding them to the tables made the tables very cumbersome. 

The text on lines 235-240 fails to mention the mean/median duration of follow-up of this retrospective cohort study. 

Response: The cohort for the study was all individuals (550,332) who belonged to the two health insurance funds at any time between 1 March 2020 and 31 December 2022. We indicate in the methods that “Although the overall study period extended from 1 March 2020 to 31 December 2022 (34 months or 2.83 years), there are some individuals who joined the health insurance funds after 1 March 2020 and some who left before 31 December 2022. For individuals who were members of the insurance funds for the entire period, the study start date and end date were recorded as their start and end dates. For those who joined the insurance funds after 1 March 2020 or left before 31 December 2022 the actual entry and exit dates were recorded as the study start or end dates.”

We have amended the text and added the average duration of membership of these individuals over the study period as an indicator of the follow-up period. 

In the discussion section, we also added text to recognise possible biases due to subjects not being part of the funds for the entire follow-up period as a limitation of the study.

What was the proportion of loss-to-follow up? 

Response: Following on the above, we have indicated in the text the proportion of individuals who were members of the funds for the entire period of the study. 

Can you complete the footnotes of Table 1 with some information on "chronic conditions" (which chronic conditions did you observe, at least the major ones? This is especially important because those chronic conditions are also predictors of COVID-19 related hospitalisations.)? 

Response: We have listed the Covid-19 comorbidities considered in the analysis in the and outcomes of interest section. These included: cancers, chronic renal disease, congestive cardiac failure, chronic obstructive pulmonary disease, diabetes mellitus, HIV, hypercholesterolaemia, hypertension, hypothyroidism, ischaemic heart disease, pregnancy and tuberculosis. We did not re-list the conditions as footnotes under Table 1 as it made the table quite cumbersome. 

It is interesting to report in the manuscript why you chose Eastern Cape as the province of reference for the statistical analysis of the association between province and COVID-19 related hospitalisation? 

Response: The Eastern Cape was the default choice as the reference group on account of it being first in the alphabet thus making it easier to reference in the table. We have not added this in the text as it should not change the interpretation. 

In the footnotes of Table 2, is it not “status at time of hospitalisation” rather than “Status at tim,e of hospital event”? 

Response: We have corrected the typo.

Should the title of Table 2 not be something like “Results of univariate and multivariate analysis of predictive factors of hospitalisation for COVID-19”? 

Response: We have reworded the titles of the tables to read: 

Table 1. Characteristics of study population 

Table 2. Univariate and multivariate analysis of impact of COVID-19 vaccination on risk of hospitalisation for COVID-19 infection

Table 3. Univariate and multivariate analysis of impact of COVID-19 vaccination on hospital utilisation for COVID-19 infection

Table 4. Univariate and multivariate analysis of impact of COVID-19 vaccination on hospital expenditure for COVID-19 infection.

Is this punctuation “?” necessary after the phrase “Chronic condition”. And once more, it would be interesting to know the chronic conditions that were associated with hospitalisation for COVID-19.

Response: We have removed the “?”. As indicated above, we have listed the chronic conditions that were considered for the analysis in the text in the section on “Exposures and outcomes of interest”. 

Would it not be better to replace “chronic conditions” by “Comorbidities” in Table 2 and “COVID-19 comorbidities” in Table 1? 

Response: We have reworded as suggested.

I think it would be more right to write “hospitalisation for COVID-19” rather than COVID-19 related hospitalisation in Table 2 and throughout the manuscript. Same remarks for Tables 3, 4, S2 and S3. 

Response: We have reworded as suggested.

Should the title of Table 1 not be “Characteristics of the study population”? 

Response: We reworded as suggested.

Is it not "versus the rest of the insured" (instead of "versus rest of the rest of the insured" in the title of Table S1? 

Response: We have corrected the wording. 

Should it not read “Hospital utilisation for COVID-19 infection” on line 263? Same remark for the sub-title on line 288. Similarly, should it not be “The overall rate of hospital utilisation for COVID-19 infection was…” on line 264? Same remark for the sentence on line 289. 

Response: We have reworded as suggested.

It would be better to report the costs in US dollars like this: $ the amount and then US, e.g., $12 US

Response: We have changed the notation of cost in US dollars to $ and specified in the methods section that the “$” refers to US dollars. 

DISCUSSION. Line 321: 7.45 per? 

Response: We have reworded the sentence.

Caution should be exercised when choosing words and interpreting findings in the discussion. For example, did you really assess risk factors (doi:10.1097/EDE.0b013e3181c1ea43.)? 

Response: We have edited and reworded to make the wording softer.

To avoid creating confusion in the discussion section, I would advise to start with a statement of main study findings (as you did) in a paragraph, discuss each of the assessed ouctomes of interest (hospitalisation due to COVID-19 infection, hospital utilization due to COVID-19 infection, and direct costs due to COVID-19 infection) in a separate paragraph (shorter than current paragraphs), and finally state study strengths, limitations and implications (for policies, clinical practice, and future research using the same insurance records and other records) in one to two paragraphs. 

Response: We have created a separate paragraph for each of the outcomes of interest and edited to reduce script length and number of references. 

When stating study limitations in the discussion section, it is very important to address the generalizability of your findings (this is a reason why you should provide us with 95% confidence intervals of the estimated rates; so, the text on lines 399-403 should be in the limitations statement.) and unavoidable study limitations related to the study design (cohort in this case; so what about selection bias, unmeasured confounding? 

Response: We have included the 95% confidence intervals for the findings of the study in the discussion section, reduced the number of other studies referred to and ensured that the 95% confidence intervals reported by the included studies are provided. We have also amended the wording to indicate that the Cox Proportional Hazards model and hazards ratios were used and why. We have reworded the text to more clearly acknowledge the limitations of the study, including the potential for selection and confounding bias and limitations regarding generalisability. 

Line 340: “in most other studies”. Which ones (you mentioned in the introduction that there is a lack of studies similar to this one)? 

Response: We have edited the sentence to remove the reference to other studies. 

Same remark on line 344 (“other studies”). And indeed, you do not report any effect magnitude estimate (RR, HR or even OR) from any of the studies cited on lines 343-360. 

Response: We have edited the sentence to remove the reference to other studies. 

If there are no studies comparable to yours (in terms of methodology) with effect magnitude estimates but only incidence estimates for hospitalization due to COVID-19 after exposure to COVID-19 vaccination, then discuss incidence rates and call for further studies to further assess effect magnitude estimates of hospitalisation due to COVID-19 infection after exposure to COVID-19 vaccination. 

Response: We have revised the text to indicate the magnitude of the effect (together with 95% CI) of the cited studies. We used standard methods to analyse data of this nature. It is standard to use Cox PH to estimate hazard ratios in the presence of censoring. It is also common to use zero-inflated models for outcomes that are a mixture of a zero and non-zero component.

Not even a small comment about patient survival when you used a Cox proportional Hazards model (why was it not even a an (secondary at least?) outcome of interest in the first place? I guess patients were hospitalised for severe/critical COVID-19 and may be hospital utilisation for COVID-19 infection included hospital utilisation for COVID-19 cases not hospitalised (those with mild to moderate COVID_19 infection) and COVID-19 cases (those with severe/critical COVID-19 infection). However, there is no such specification in any part of the article not even in the discussion when addressing limitations. If the information on COVID-19 severity is available, it would also be great to specify which COVID-19 infection severity classification scheme was used in you

---

## [Editor Report · Decision Letter 2]

3 Jan 2025

Impact of COVID-19 vaccination on hospitalization, hospital utilization and expenditure for COVID-19:  a retrospective cohort analysis of a South African private health insured population.

PONE-D-24-26424R2

Dear Dr. Solanki,

We’re pleased to inform you that your manuscript has been judged scientifically suitable for publication and will be formally accepted for publication once it meets all outstanding technical requirements.

Kind regards,

Mickael Essouma, M. D.

Academic Editor

PLOS ONE

Additional Editor Comments (optional):

Congratulations to the authors for this high-quality retrospective research and for demonstrating intellectual humility throughout the discussion section.

I have some last minute edits in the attached pdf.
---

## [Editor Report · Acceptance letter]

16 Jan 2025

PONE-D-24-26424R2 

PLOS ONE

Dear Dr. Solanki, 

I'm pleased to inform you that your manuscript has been deemed suitable for publication in PLOS ONE. Congratulations! Your manuscript is now being handed over to our production team.

Kind regards, 

on behalf of

Dr. Mickael Essouma 

Academic Editor

PLOS ONE